# Preparation and Characterization of Electrospun Double-layered Nanocomposites Membranes as a Carrier for *Centella asiatica* (L.)

**DOI:** 10.3390/polym12112653

**Published:** 2020-11-11

**Authors:** Cláudia Mouro, Raul Fangueiro, Isabel C. Gouveia

**Affiliations:** 1FibEnTech Research Unit, Faculty of Engineering, University of Beira Interior, 6201-001 Covilhã, Portugal; d1684@ubi.pt; 2Centre for Textile Science and Technology (2C2T), University of Minho, 4710-057 Guimarães, Portugal; rfangueiro@dem.uminho.pt

**Keywords:** polycaprolactone, chitosan, sodium tripolyphosphate, poly(vinyl alcohol), *Centella asiatica*, double-layered nanocomposite membranes

## Abstract

A wide range of naturally derived and synthetic biodegradable and biocompatible polymers are today regarded as promising materials for improving skin regeneration. Alongside this, these materials have been explored in conjunction with different types of antimicrobial and bioactive agents, especially natural-derived compounds, to enhance their biological properties. Herein, a double-layered nanocomposite dressing membrane was fabricated with two distinct layers. A bottom layer from Chitosan-Sodium tripolyphosphate (CS-TPP) and Poly(vinyl alcohol) (PVA) containing *Centella asiatica* (L.) (CA) was electrospun directly over a Polycaprolactone (PCL) layer to improve the biologic performance of the electrospun nanofibers. In turn, the PCL layer was designed to provide mechanical support to the damaged tissue. The results revealed that the produced double-layered nanocomposite membrane closely resembles the mechanical, porosity, and wettability features required for skin tissue engineering. On the other hand, the in vitro drug release profile of the PCL/PVA_CS-TPP containing CA exhibited a controlled release for 10 days. Moreover, the PVA_CS-TPP_CA’s bottom layer displayed the highest antibacterial activity against *Staphylococcus aureus* (*S. aureus*) (99.96 ± 6.04%) and *Pseudomonas aeruginosa* (*P. aeruginosa*) (99.94 ± 0.67%), which is responsible for avoiding bacterial penetration while endowing bioactive properties. Finally, the 3-(4,5-Dimethyl-2-thiazolyl)-2,5-diphenyl-2H-tetrazolium bromide (MTT) assay showed that this nanocomposite membrane was not cytotoxic for normal human dermal fibroblasts (NHDF) cells. Therefore, these findings suggest the potential use of the double-layered PCL/PVA_CS-TPP_CA as an efficient bionanocomposite dressing material.

## 1. Introduction

Wound healing is a dynamic and complex process that requires cellular interactions between a wide variety of cell types [1,2,3]. These interactions are mediated through a coordinated cascade of biological events synergistically controlled by numerous bioactive molecules, such as growth factors, anti-inflammatory agents, and vitamins. However, the physiology of the healing process may be impaired by several factors [1,2,3]. Among them, bacterial colonization and subsequent infections remain one of the most serious complications after skin structure is compromised [1,4,5].

Generally, when pathogenic microorganisms contaminate skin wounds, the immune system mobilizes its energy trying to suppress the invasion of these pathogens instead of focusing on the re-establishment of the native skin’s structural and functional features [1,4,5,6]. If infection occurs, microorganisms, particularly bacteria, can produce endotoxins that stimulate the expression of pro-inflammatory cytokines and encourage an extended inflammatory response. In this case, wounds exhibit increased levels of metalloproteinases (MMPs), which provide an unsuitable environment for the production of new skin’s extracellular matrix (ECM) components, delaying or even interrupting the healing process [1,5,6].

Concerning this, several wound dressings displaying antimicrobial properties have been developed to protect the wound from infection and enhance the healing process. Nonetheless, it is essential to improve their performance to suppress this health problem and reduce the occurrence of life-threatening complications [1,5,6,7]. For this purpose, different combinations of both synthetic and natural biocompatible and biodegradable polymers have been explored to produce 3D nanofibrous membranes that mimic the architecture of the skin’s ECM [1,6,7,8].

So far, to successfully produce nanofibers as potential wound dressings, several techniques have been used, namely self-assembly, phase separation, drawing, template synthesis, and electrospinning. Electrospinning has been considered as one of the most efficient, versatile, and cost-effective methods to produce nanocomposite dressing materials with the desirable features [1,7].

The unique structural and morphological properties of the electrospun nanofibers, like the high specific area to volume ratio, interconnected pores, and the smaller fiber diameters, closely resemble the structure of collagen fibers found in the normal skin’s ECM. Alongside this, their porous structure can support cell adhesion, migration, and proliferation, and regulate the release of both growth factors and signaling molecules, which are required to achieve wound healing [1,7,8,9,10,11]. The electrospun nanofibers can also provide quick hemostasis, enhance exudate absorption, nutrients, and gas permeability, as well as preventing bacterial penetration and colonization. Moreover, electrospinning provides the operational ability for incorporating antimicrobial and/or bioactive agents, which enable the improvement of the biologic behavior of these wound dressing materials [1,7,8,9].

To accomplish that, several different approaches have been used as an alternative to traditional single-fluid electrospinning [12]. Among them, double-fluid and multiple-fluid electrospinning methods have been extensively studied to produce core–shell nanofibers and provide specific release profiles [13,14]. In the coaxial electrospinning system, namely in modified coaxial electrospinning processes, different coaxial spinneret needles have been designed [15]. Furthermore, two needles side-by-side have been applied to produce nanofibers with Janus structures [14,16,17]. On the other hand, multiple-fluid systems with distinct spinneret arrangements like traditional and modified triaxial spinnerets and electrospun nanofibers with a common shell and two separate cores have been developed to ensure a sustained release of the incorporated agents [14,18,19,20,21]. However, those new methods for manipulating the inner chamber structure are complicated. Thus, traditional single-fluid blending electrospinning is still the mainstream method due to its straightforward operation, easy scale-up, and remarkable power for tailoring the components and compositions of resultant composite nanofibers. Additionally, manipulation of deposition on the collector with different electrospun nanofibers has been studied in recent works and is being further explored to improve the functional performances of the electrospun wound dressing materials containing antimicrobial agents [11,22,23,24].

Among the different antimicrobial agents incorporated so far in electrospun wound dressings, antibiotics and nanoparticles have been widely explored due to their capability to avoid bacteria penetration and colonization into the wound site [1,5,7,9]. Nonetheless, the growing threat of antibiotic resistance and their toxicity have encouraged the use of natural products to avoid bacterial contamination. Regarding that, compounds obtained from natural sources, like medicinal plants, have been regarded as a powerful natural supplement for the management and treatment of wounds [1,7,9,25,26]: mainly crude plant extracts, which are ecologically sustainable mixtures rich in interesting bioactive phytochemicals, such as tannins, alkaloids, carbohydrates and glycosides, terpenoids, steroids, flavonoids, and coumarins with multiple healing benefits [7,9,26,27].

Herein, we produced a novel bionanocomposite dressing membrane with a double-layered structure through electrospinning. Polycaprolactone (PCL), a hydrophobic synthetic polymer, was used as the main component of the first layer, due to its biocompatibility, desirable mechanical strength, and ability to act as a protective barrier [10,28,29,30]. On the other hand, the second layer of Poly(vinyl alcohol) (PVA) and Chitosan-Sodium tripolyphosphate (CS-TPP) containing *Centella asiatica* (L.) (CA) was designed to be in direct contact with the injured skin and enhance the healing process [30,31].

CA is a member of the *Apiaceae* family, and it has been widely used for the treatment of dermatoses, skin lesions such as burns, excoriations, hypertrophic scars or eczema, and other skin diseases, like leprosy and psoriasis, as well as in non-dermatologic conditions. CA displays different terpenoids, known as centelloids, including asiaticoside, madecassoside, centelloside, centellose, brahminoside, thankunizide, sceffoleoside, brahmoside, and asiatic, centellic, brahmic, and madecassic acids which are responsible for conferring several therapeutic properties to CA [32,33,34]. Moreover, the extracts obtained from this medicinal plant are known for their capability to stimulate fibroblast proliferation, collagen synthesis, and angiogenesis [32]. In turn, Chitosan (CS), one of the most abundant natural polysaccharides, is known for its ability to stimulate collagen synthesis, as well as bactericidal and hemostatic properties. Alongside this, CS possesses amine functional groups on its backbone chains, which in acidic aqueous media ensure a high density of positive charges [31,35,36,37]. Thus, under these conditions, CS can be ionically cross-linked with biodegradable and biocompatible polyanions, as the Sodium tripolyphosphate (TPP), forming polyelectrolyte complexes as antimicrobial and/or bioactive agents delivery carriers [31,35]. Nevertheless, the CS-TPP solutions are difficult and unstable to electrospun into a fibrous structure, due to the high viscosity of the CS at low pH values [31,37,38,39]. To overcome this limitation, PVA, one of the most commonly used water-soluble synthetic polymers, was added to the CS-TPP blend to enhance fiber-forming ability [31,36,38,39].

Hence, in this work, we aimed to take advantage of the capability offered by double-layered PCL/PVA_CS-TPP_CA to improve the wound healing process, namely the benefits that the ionically cross-linked electrospun PVA_CS-TPP nanofibers display to control the release of the crude CA extract according to the demands for antimicrobial wound care products.

## 2. Materials and Methods

### 2.1. Materials

*Centella asiatica* (CA) was supplied from a Portuguese botanic shop (CHÁ HUNOS, Lda., Portugal) without any additives. Normal human dermal fibroblasts (NHDF) cells were purchased from ATCC—American Type Culture Collection. Polycaprolactone (PCL) (MW 80.000 g/mol), Chitosan (CS) (low molecular weight) were acquired from Sigma-Aldrich. Polyvinyl Alcohol (PVA) (MW 115.000 g/mol) was purchased from VWR Chemicals. Ethanol absolute, Chloroform, Dimethylformamide (DMF), and Glacial acetic acid were purchased from Fisher Chemical. Nutrient agar (NA), Nutrient broth (NB), and Agar for microbiology were provided from Fluka. Brain Heart Infusion (BHI) broth was obtained from Panreac. Mueller Hinton broth (MHB), Tween 80, Sodium Hydroxide (NaOH), Sodium Chloride (NaCl), Trypsin, and 3-(4,5-Dimethyl-2-thiazolyl)-2,5-diphenyl-2H-tetrazolium bromide (MTT) were bought from Sigma-Aldrich. Phosphate-buffered saline (PBS) and Sodium tripolyphosphate (TPP) were purchased from Alfa Aesar. All solvents were used as received from the manufacturer.

### 2.2. Ethanol Extraction of Crude Centella Asiatica (CA) Plant

The dried and powdered aerial plant parts (4 g) were macerated using 40 mL of 95% ethanol as solvent at room temperature for 24 h. Then, the supernatant from the ethanol extraction was directly filtered through Whatman filter paper, and after that, the acquired filtrate was dried under reduced pressure to obtain dry CA extract. The yield of the fresh plant was 18.81% (dry weight of the extract obtained after solvent removal per weight of plant) (*w*/*w*). Finally, the crude CA extract was stored following good storage practices and next re-suspended in 45% (*v*/*v*) ethanol for further experiments.

### 2.3. Minimum Inhibitory Concentration (MIC) of the Crude CA extract

Minimum inhibitory concentration (MIC) of the crude CA extract was assessed against *Staphylococcus aureus* (ATTC 6538) (*S. aureus*) and *Pseudomonas aeruginosa* (PA25) (*P. aeruginosa*) using the broth microdilution method on 96 multi-well polystyrene plates (Sigma-Aldrich), according to the CLSI M07-A6 document. Briefly, serial dilutions of crude CA extract were prepared in sterile Mueller Hinton Broth (MHB) to obtain the desired extract concentrations (between 50 and 0.15 mg/mL). Then, 50 µL of each CA dilution containing 50 µL of a bacterial suspension (adjusted to ~10^7^ CFU/mL in MHB) was applied to the microplate wells in triplicate. The plates were after that incubated at 37 °C for 18–24 h. The MIC was defined as the lowest concentration of the crude CA extract at which there was no visible growth of *S. aureus* and *P. aeruginosa* (no solution turbidity on naked eyes). MHB with bacterial suspensions was added as a positive control, whereas only MHB was used as a negative control.

### 2.4. Fabrication of the Double-Layered Nanocomposites Membranes

The double-layered nanocomposites membranes were fabricated using the Nanospider technology (Nanospider laboratory machine NS LAB 500S from Elmarco S.R.O., Czech Republic, http://www.elmarco.com), as a modified electrospinning method.

Top layer: Initially, a PCL solution (8% PCL (*w*/*v*)) was prepared in chloroform/DMF at 30:20 volume ratio. The resultant solution was electrospun at 75.0 kV, using a working distance of 15 cm and an electrode rotation rate of 55 Hz (electrode spin = 8.8 r/min).

Bottom layer: CS-TPP blend was prepared according to a previously reported method by Nguyen et al. [40] using sodium tripolyphosphate (TPP) as a crosslinking agent. Briefly, 0.2% (*w*/*v*) chitosan was dissolved in 0.35% acetic acid and kept overnight at room temperature. The pH of the resulting chitosan solution was then adjusted to pH 5.5 using a 0.5 M sodium hydroxide (NaOH) solution. In turn, a TPP solution was prepared in distilled water at a concentration of 0.25% (*w*/*v*). The CS-TPP was produced by dropping the TPP solution into the CS solution under vigorous stirring in a volume ratio of 6:1 for 60 min at room temperature. Afterward, the CS-TPP was blended with 10% (*w*/*v*) of Polyvinyl Alcohol (PVA) dissolved in distilled water at 90 °C with a volume ratio of 70:30, respectively. Additionally, the crude CA extract incorporation into CS-TPP was achieved by adding 3 mg/mL of the CA extract in the TPP solution. The CS-TPP_CA was produced following the same procedure as for the CS-TPP.

After polymer solutions were obtained, they were placed in a container with a rotating spinning electrode, and electrospun on top of the recently prepared PCL’s top layer at an electrode spin of 45 Hz (electrode spin = 7.2 r/min), using a working distance of 15 cm and an applied voltage of 75 kV.

Finally, the fabricated double-layered nanocomposites membranes (PCL/PVA_CS-TPP and PCL/PVA_CS-TPP_CA) were characterized through in vitro assays to assess their appropriateness as a wound dressing material.

### 2.5. Characterization of the Produced Double-Layered Nanocomposites Membranes

#### 2.5.1. Scanning Electron Microscopy (SEM) Imaging and Analysis

The surface morphology of the electrospun nanofibers of the top layer (PCL) and bottom layers (PVA_CS-TPP and PVA_CS-TPP_CA) of the developed double-layered nanocomposite membranes was observed using scanning electron microscopy (SEM) (S2700, Hitachi, Tokyo, Japan) at an accelerating voltage of 20 kV. First, the samples were mounted on aluminum stubs and sputter-coated with a thin gold layer in an Emitech K550 sputter coater (Quorum Technologies Ltd., Laughton, East Sussex, UK) for better conductivity during imaging. The fiber diameters were measured from the obtained SEM images using ImageJ software (National Institutes of Health, MD, USA) and the size-frequency distributions constructed with GraphPad Prism 6 software (GraphPad Software, La Jolla, CA, USA).

#### 2.5.2. Attenuated Total Reflectance–Fourier Transform Infrared Spectroscopy Study

The chemical composition of the top layer (PCL), the bottom layers (PVA_CS-TPP and PVA_CS-TPP_CA), and their raw materials was analyzed using attenuated total reflectance–Fourier transform infrared spectroscopy (ATR–FTIR, Thermo-Nicolet is10 FT-IR Spectrophotometer, Waltham, MA, USA). The spectra of the samples were recorded in a spectral width ranging from 400–4000 cm^−1^ with an average of 32 scans min^−1^ and a spectral resolution of 4 cm^−1^.

#### 2.5.3. Differential Scanning Calorimetry (DSC)

The thermal behavior of the PCL’s top layer and the bottom layers of PVA_CS-TPP with and without CA extract was evaluated by differential scanning calorimetry (DSC) (DSC 204 Phoenix Netzsch, Germany). Briefly, about 5 mg of each sample was filled in small aluminum containers, and the non-isothermal scans performed from 30 °C to 200 °C at a heating rate of 5 °C/min, with a nitrogen-replacing atmosphere.

#### 2.5.4. Assessment of the Mechanical Characteristics of the Produced Double-Layered Nanocomposites Membranes

The tensile test was carried out in dry conditions according to the ASTM standard D3039/D3039M to evaluate the mechanical characteristics of the produced double-layered nanocomposites membranes. Briefly, samples of the PCL/PVA_CS-TPP and PCL/PVA_CS-TPP containing CA samples (n = 5) were cut into rectangular strips of 1 cm × 4 cm, and then the thickness was measured with a micrometer (Adamel Lhomargy MI20, France). The tensile test was performed using a dynamometer (DY-35 Adamel Lhomargy, France) by using a load cell of 10-N. The samples were mounted vertically between the clamps of the tensile tester, and a speed of 2 mm/min used until the membranes were ruptured. Finally, the tensile strength, Young’s modulus, and elongation at break were determined.

#### 2.5.5. Measurement of the Total Porosity

The total porosity of the dried PCL’s top layer and bottom layers of PVA_CS-TPP and PVA_CS-TPP_CA was measured using a fluid displacement method and conducted as previously described by Yeh et al. [41]. Absolute ethanol with density $\rho_{\varepsilon }$ was used as displacement liquid because it can easily penetrate the porous structure without inducing negligible shrinking or swelling as a non-solvent of both layers. Briefly, a graduated cylinder with ethanol was weighed (W_1_), then a dried sample with a known weight (W_s_) was immersed into the cylinder containing the displacement liquid. After that, this assembly was placed in an ultrasonic bath (Ultrasons-H, P-Selecta) for 40 min at 30 °C. After this period, the volume of ethanol in the graduated cylinder was refilled and weighed as W_2_. The sample saturated with ethanol was taken out from the cylinder, and its weight determined as W_3_. The porosity (ε) of both layers was estimated through the following (Equations (1)–(3)):(1)$$V_{s}=\frac{\left(W_{1}-W_{2}+W_{s}\right)}{\rho_{\varepsilon }}$$
(2)$$V_{p}=\frac{\left(W_{2}-W_{3}-W_{s}\right)}{\rho_{\varepsilon }}$$
(3)$$\varepsilon \;\left(\%\right)=\frac{V_{p}}{\left(V_{p}+V_{s}\right)}\times 100\;\Leftrightarrow \varepsilon \;\left(\%\right)=\frac{\left(W_{2}-W_{3}-W_{s}\right)}{\left(W_{1}-W_{3}\right)}\times 100$$
where $V_{s}$ is the volume of the sample, and $V_{p}$ is the volume of the sample pores. For each sample, the porosity measurements were performed in triplicate, and the average ± standard deviation (S.D.) shown for each sample.

#### 2.5.6. Evaluation of Wettability Properties

The water contact angles (WCA) at the surface of both layers (PCL, PVA_CS-TPP, and PVA_CS-TPP_CA) were determined using a Data Physics Contact Angle Goniometer (OCAH-200) for surface-wetting characterization. Briefly, each sample was placed on the measuring stage, then water drops (4 µL) were seated onto the surface of the samples at different locations at 25 °C. The reported WCA values were the average of at least three independent measurements (n = 3).

#### 2.5.7. Analysis of the In Vitro Swelling Behavior

The swelling degree of the produced double-layered nanocomposites membranes (PCL/PVA_CS-TPP and PCL/PVA_CS-TPP_CA) was investigated in a phosphate buffer solution (PBS) at a pH of 5.5 by using a gravimetric method. Briefly, the pre-weighted dry samples (W_0_) were immersed in the PBS at 37 °C. At specific time points, the swollen samples were removed from the PBS buffer solution and reweighted after kindly wiping the excess buffer off the samples (W_t_). All measurements were performed in triplicate (n = 3) and the amount of water uptake determined according to the following Equation (4):(4)$$\mathrm{Swelling}\;\mathrm{Ratio}\;\left(\%\right)=\frac{\left(W_{t}-W_{0}\right)}{W_{0}}\times 100$$

#### 2.5.8. Study of the In Vitro Biodegradation Profile

The physical integrity behaviors were analyzed from the weight loss of the produced double-layered nanocomposites membranes. Briefly, the dried samples with the initial weight of (W_0_) were immersed into PBS solution (pH = 5.5) at 37 °C. At predetermined time intervals (1, 4, 7, and 10 days), the samples (n = 3) were removed from the PBS solution, rinsed with distilled water to remove residual buffer salts, oven-dried, and reweighted (W_d_). Finally, the weight loss (%) of each sample was determined based on Equation (5):(5)$$\mathrm{Weight}\;\mathrm{loss}\;\left(\%\right)=\frac{\left(W_{0}-W_{d}\right)}{W_{d}}\times 100$$

#### 2.5.9. Water Vapor Transmission Rate (WVTR) Analysis

The gravimetric assay based on the ASTM E96/E96M-15 standard was used to evaluate the water vapor transmission rate (WVTR) of the produced double-layered nanocomposites membranes. Briefly, sample circles (1.2 cm diameter) were cut and carefully attached to the mouths of test tubes containing 10 mL of deionized water. The circular opening of the test tubes was sealed using parafilm, and the samples–glass tubes assembly placed in an incubator at 37 °C. At predetermined intervals, the amount of water evaporation was estimated by the changes in their weight over time. The WVTR was calculated according to Equation (6):(6)$$\mathrm{Water}\;\mathrm{vapor}\;\mathrm{transmission}\;\mathrm{rate}\;\left(\mathrm{WVTR}\right)=\frac{W_{\mathrm{loss}}}{A}\;\left(g/m^{2}/\mathrm{day}\right)$$
where W_loss_ is the daily weight loss of water and A is the test area in m^2^.

### 2.6. Analysis of the In Vitro CA Release from Double-Layered Nanocomposites Membranes

The in vitro release profile of the double-layered nanocomposites membranes containing crude CA extract was investigated in PBS (pH = 5.5) containing 10% (*v*/*v*) of ethanol. The amount of released CA in PBS was monitored by a UV–Vis spectrophotometer at a wavelength of 370 nm [42]. Briefly, the double-layered PCL/PVA_CS-TPP_CA membranes were kept immersed in PBS buffer at 37 °C and 100 rpm for 10 days. At specific time points, a fixed volume of released medium was taken out from the incubation medium, and an equal amount of fresh buffer solution refilled to maintain the sink condition. The amount of crude CA extract released was measured by converting its detected UV absorbance to its concentration according to the calibration curve constructed from a series of CA standard solutions (from 0.00 mg/mL to 5.00 mg/mL). After that, the data obtained were evaluated to determine the cumulative percentage of the released CA from the samples at each immersion time point. The experiments were conducted in triplicate (n = 3).

### 2.7. Assessment of the Antibacterial Properties of the Produced Double-Layered Nanocomposites Membranes

The antibacterial activity of both layers (PCL, PVA_CS-TPP, and PVA_CS-TPP_CA) was exanimated against *S. aureus* and *P. aeruginosa* following the guidelines established by the Standard Test Method for Determining the Activity of Incorporated Antimicrobial Agent(s) in Polymeric or Hydrophobic Materials (ASTM E2180-07 standard). Firstly, *S. aureus* and *P. aeruginosa* were cultivated in nutrient broth (NB) and brain–heart infusion broth (BHI) in a shaking incubator at 37 °C and 110 rpm for 18–24 h, respectively. After that, the bacterial suspensions were diluted until the bacterial concentration reached ~10^8^ CFU/mL, then added to the previously prepared agar slurry to facilitate surface interaction. A thin layer of inoculated agar slurry was pipetted onto the samples and then left to gel at room temperature before incubation at 37 °C for 18–24 h. The surviving bacteria were analyzed immediately (T_0h_) and after incubation (T_24h_) by elution of the agar slurry inoculum from the test samples. After bacteria elution, serial dilutions were made in NaCl and pipetted on agar plates, and incubated at 37 °C for 18–24 h. Finally, the number of surviving colonies following incubation was counted, and the counts used to establish the log (CFU/mL).

### 2.8. Analysis of the In Vitro Cell Viability

The cytotoxicity of the double-layered nanocomposites membranes (PCL/PVA_CS-TPP and PCL/PVA_CS-TPP_CA) was evaluated through colorimetric 3-(4,5-Dimethyl-2-thiazolyl)-2,5-diphenyl-2H-tetrazolium bromide (MTT) assay according to ISO 10993–5 (Biological evaluation of medical devices–Part 5: Tests for in vitro cytotoxicity). Firstly, the normal human dermal fibroblasts (NHDF) cells were cultured in a medium supplemented with fetal bovine serum (FBS) in a humidified incubator at 37 °C under a 5% CO_2_ atmosphere. Afterward, the samples cut into round disks (with a diameter of ~6 mm) were placed at the center of each well in 24-well plates, then sterilized by UV irradiation for 1 h before cell seeding. After that, 1 × 10^4^ cells/well were seeded in each well containing the sterilized membranes and incubated with 5% CO_2_ at 37 °C for 1, 3, and 7 days. During these intervals of time, the medium was removed, and a mixture of fresh culture medium with the MTT reagent added to each well. After being incubated for 4 h under the same conditions, the content of each well was again removed and replaced by DMSO to dissolve the formazan crystals. Finally, the absorbance of each membrane was measured at 570 nm using a spectrophotometric plate reader (Biorad xMark microplate spectrophotometer). Cells incubated without samples (K^−^) and cells with EtOH (96%) (K^+^) were chosen as control groups. The positive control (K^+^) was added in separate 24 well plates to avoid false results caused by EtOH (96%).

### 2.9. Statistical Analysis

Statistical analysis was performed from the one-way ANOVA, followed by multiple comparison test Turkey using GraphPad Prism 6 software (GraphPad Software, La Jolla, CA, USA) with a statistical significance of *p* < 0.05.

## 3. Results and Discussion

### 3.1. Minimal Inhibitory Concentration (MIC) of the Crude CA Extract

The antimicrobial susceptibility of the crude CA extract was determined by the MIC. The MIC value against *S. aureus* was found to be 1.40 mg/mL, while the value for *P. aeruginosa* was 2.80 mg/mL. These values were lower than those obtained by Yao et al. [43], who revealed MIC values to the ethanolic extract of CA of 6.25 mg/mL and 25 mg/mL against *S. aureus* and *P. aeruginosa*, respectively. These results proved that the antibacterial activity of the medicinal plants depends on the specific active compounds present in the extract.

### 3.2. Characterization of the Produced Double-Layered Nanocomposites Membranes

#### 3.2.1. Scanning Electron Microscopy (SEM) Imaging and Analysis

In this study, the surface morphologies and diameter distributions of the electrospun nanofibers from PCL’s top layer and the bottom layers of PVA_CS-TPP and PVA_CS-TPP incorporated with the crude CA extract, respectively, are displayed in Figure 1a. The SEM images show that both layers exhibit a random distribution of nanofibers with interconnected pores. The average diameters of the smooth PCL structure were determined to be 277.63 ± 85.19 nm, which is in agreement with other studies performed with PCL [44]. In turn, the average fiber diameter of the smooth and bead-free structures of PVA_CS-TPP was decreased from 323.85 ± 91.07 nm to 284.34 ± 75.79 nm when the crude CA extract was incorporated, as a result of reduction of the viscosity of the electrospinning solution. In this way, these results suggest that the produced double-layered nanocomposites membranes resemble the fibrous morphology and architecture of the natural extracellular matrix (ECM) since the nanofibers exhibit diameters within the size range of the collagen fibers of ECM (50–400 nm), being able to promote cell adhesion and proliferation [45,46].

According to the cross-sectional image, Figure 1b, it is possible to observe the two different layers of the produced double-layered nanocomposites membranes.

#### 3.2.2. Attenuated Total Reflectance-Fourier Transform Infrared Spectroscopy Study

The acquired ATR-FTIR spectra of the produced double-layered nanocomposites membranes are presented in Figure 2. The spectrum of the PCL’s top layer displays its characteristic bands, Figure 2a. The peaks at 2865.22 and 2943.10 cm^−1^ belongs to the symmetric and asymmetric CH_2_ stretching vibration, while the band at 1722.95 cm^−1^ corresponds to the C=O stretching vibration [47].

In turn, the spectrum of the PVA_CS-TPP’s bottom layer shows the characteristic peaks of the PVA and CS at 3316.14 and 2936.47 cm^−1^, attributed to the O-H and CH_2_ stretching vibration, respectively, and a peak at 1642.57 cm^−1^ assigned to C=O stretching of a primary amide, Figure 2b. These bands revealed that the PVA and CS-TPP were successfully dispersed in the nanofibers [31]. Moreover, when the CA extract was incorporated into the PVA_CS-TPP nanofibers, a higher intensity of the peaks was observed once the characteristic peaks of the crude CA extract overlapped with the bands of PVA_CS-TPP, Figure 2b. A similar effect was previously reported by Amina et al. [48], who showed that the characteristic peaks of PU nanofibers overlapped with the bands of aqueous extract of *Grewia mollis* (*G. mollis*), leading to a higher intensity in the PU/*G. mollis* nanofibers’ spectrum. However, the spectrum of raw CA confirms its characteristics peaks at 3322.12 cm^−1^ (O-H stretching vibration of carboxylic acid group), 1656.07 cm^−1^ (C-O stretching vibration), 1451.17 cm^−1^ (C-H in-plane bending vibration), 1375.48 cm^−1^ (C-N stretching vibration, aromatic amide), and 1024.98 cm^−1^ (C-O stretching) [49].

#### 3.2.3. Differential Scanning Calorimetry (DSC)

The thermal properties of both layers (PCL’s top layer and the bottom layers of PVA_CS-TPP and PVA_CS-TPP containing crude CA extract) were evaluated by DSC, as demonstrated in Figure 3.

In PCL’s top layer, the endothermic peak at 62.96 °C corresponds to the melting temperature (T_m_) of PCL. This result is in agreement with the data available in the literature for electrospun PCL membranes (T_m_(PCL) = 60.10 °C) [50].

On the other hand, the raw PVA_CS with the PVA_CS-TPP nanofibers display an endothermic peak at 62.24 °C corresponding to the evaporation of water and acetic acid solvents [51,52]. In addition, a weak endothermic peak was found at 188.43 °C, which is due to the melting of PVA crystals [51]. Additionally, the DSC thermograms suggest that the presence of CS-TPP shifted the endothermic peaks to a higher temperature, confirming the thermal stability of the ionically cross-linked electrospun PVA_CS-TPP nanofibers. Moreover, the incorporation of the crude CA extract slightly changed the T_m_ of the PVA_CS-TPP nanofibers, indicating that the incorporation of the crude plant extract slightly enhance the thermal properties of the bottom layer of the produced double-layered nanocomposite membrane. Therefore, the electrospun PVA_CS-TPP_CA nanofibers were revealed to be thermally stable to support cell growth, which is essential for improving the healing process.

Similarly, Ghaseminezhad et al. [53] developed electrospun PCL/Gelatin nanofibers loaded with different amounts of Althea officinalis (AO) and demonstrated that till 15 wt % AO the thermal stability of these electrospun nanofibers was slightly improved.

#### 3.2.4. Assessment of the Mechanical Characteristics of the Produced Electrospun Double-Layered Nanocomposites Membranes

The mechanical characteristics, like tension, elasticity, and Young’s modulus, of the produced double-layered nanocomposites membranes, are summarized in Table 1. The electrospun PCL/PVA_CS-TPP membrane exhibited a tensile strength of 3.96 ± 0.99 MPa, Young’s modulus of 38.16 ± 5.32 MPa, and an elongation at break of 10.39 ± 1.89%. However, the mechanical strength of the nanocomposites membranes was decreased after incorporation of the crude CA extract. The tensile strength and Young’s modulus of the PCL/PVA_CS-TPP_CA were reduced to 3.03 ± 0.67 MPa and 36.36 ± 7.29 MPa, respectively. The crude CA extract also had a slight effect on the elongation at break (8.31 ± 0.61%). Nevertheless, the obtained values are close to those exhibited by native human skin, Table 1.

According to what was described above, several studies have shown that the incorporation of the crude plant extracts into electrospun nanofibers decreases the mechanical strength of the membranes, due to its weaker mechanical properties [11,54]. In turn, synthetic polymers like PCL have exhibited excellent mechanical performance [55].

Thereby, the two layers of the double-layered nanocomposites membranes attached will provide good resilience and compliance to cover a wound area, resulting in effective wound healing with minimal scarring.

#### 3.2.5. Measurement of the Total Porosity

The wound dressings’ porosity is a very important parameter for an effective healing process to occur. Herein, the porosity of the PCL’s top layer was found to be 64.01 ± 10.61%, Figure 4a. This value is in accordance with other studies that used the lowest porosity of the PCL as a protective layer against bacteria penetration [11,56]. On the other hand, the bottom layers of the PVA_CS-TPP and PVA_CS-TPP_CA displayed porosities of 92.92 ± 1.16% and 96.88 ± 1.14%, respectively, Figure 4a. The data obtained reveal that the porosity values of the bottom layers, which play a vital role in the healing process, are in the preferred range of 60 to 90% for cell adhesion, migration, and proliferation. In addition, a highly porous structure is beneficial for the improvement of the production of the new ECM components [7].

#### 3.2.6. Evaluation of Wettability Properties

The wettability of the wound dressing materials is a crucial parameter that affects their biological performance [57]. Herein, the surface wettability of both layers of the produced double-layered nanocomposite membranes was investigated from the WCA between the surface material and water droplets using the sessile drop technique. The PCL’s top layer of nanocomposites membranes exhibited a WCA value of 105.93 ± 18.85°, confirming its hydrophobic character conferred by PCL aliphatic chains, Figure 4b. On the other hand, the bottom layers of the PVA_CS-TPP, which is in contact with the wound, showed a WCA value of 50.83 ± 13.97°. This value decreased to 42.50 ± 16.93° when the crude CA extract was incorporated in this layer (PVA_CS-TPP_CA), Figure 4b. Such behavior is associated with the hydrophilic character of the plant components. Therefore, the moderate wettability of the PVA_CS-TPP and PVA_CS-TPP_CA nanofibers is in accordance with the literature since WCA values between 40° and 70° could lead to better cell adhesion, migration, and proliferation. Furthermore, values within this range are more prone to provide moist environments, enhancing the healing process, than the hydrophobic (WCA > 90°) or super hydrophilic (WCA < 20°) surfaces, as previously mentioned in the literature [56,57].

#### 3.2.7. Analysis of the In Vitro Swelling Behavior

The water absorption capability of the produced electrospun double-layered nanocomposites membranes plays an important role in the absorption of wound exudates providing a moist wound environment [58]. The swelling behavior might contribute to avoiding an extreme level of the moisture at the wound surface, and the dryness of the wound, which can lead to a delay in healing [58].

The swelling profiles of the PCL/PVA_CS-TPP and PCL/PVA_CS-TPP_CA membranes were evaluated after immersion in PBS at pH = 5.5 for 10 days to simulate the acidic environment at the wound site, Figure 4c. The electrospun PCL/PVA_CS-TPP membrane containing crude CA extract displayed a higher water absorption capability (~600%) than the PCL/PVA_CS-TPP membrane (~400%). Thus, the results suggest that the incorporation of CA extract in the bottom layer of PVA_CS-TPP provided hydrophilicity to the polymer blend, contributing to a higher amount of water retained in the interconnected fibrous pores and increasing the water absorption ratio. A similar effect was previously described by Yousefi et al. [35], who found a higher water-uptake ability when Henna extracts were incorporated into CS/Poly(ethylene oxide) (PEO) nanofibrous mats. Regarding that, the CS/PEO/Henna extract nanofibrous mats were demonstrated to exhibit the capability to efficiently remove the exudate from the wound, adjusting the wound moisture.

#### 3.2.8. Study of the In Vitro Biodegradation Profile

An ideal wound dressing must display suitable biodegradability and a degradation profile consistent with the wound repair and regeneration [59]. Herein, the electrospun double-layered nanocomposites membranes were fabricated from biodegradable materials, and the percentages of weight loss are presented in Figure 4d. The degradation studies showed that the PCL/PVA_CS-TPP exhibited a weight loss of 37.10 ± 2.33%, while the PCL/PVA_CS-TPP_CA undergo a weight loss of 48.04 ± 1.82%, respectively. Therefore, the hydrophilic character of crude CA extract contributed to both increases the water absorption and a slightly faster weight loss of the nanofibers. In addition, the PVA and CS present in the bottom layer are characterized by suffering higher weight losses than the PCL’s top layer, which displays a slow in vitro degradation profile [11].

#### 3.2.9. Water Vapor Transmission Rate (WVTR) Analysis

The membrane’s permeability is another important parameter which can ensure suitable oxygenation of the wound environment and stimulate new blood vessel, sustaining the complex cellular events during the healing process [60]. In addition, the wound dressing membranes should avoid excessive dehydration as well as the build-up of exudates [7]. The data available in the literature demonstrated that the ideal WVTR value for wound dressing materials should be between 2000–2500 g/m^2^/day [7]. Herein, a WVTR of 1162.94 ± 116.10 g/m^2^/day and 1757.12 ± 67.69 g/m^2^/day were measured for PCL/PVA_CS-TPP and PCL/PVA_CS-TPP_CA, respectively. These values appeared to be outside the range recommended for an ideal dressing. However, the WVTR values obtained herein are within the range displayed by commercial wound dressings (426–2047 g/m^2^/day) [61].

### 3.3. Analysis of the In Vitro CA Release from Electrospun Double-Layered Nanocomposites Membranes

Nowadays, loading bioactive agents into the electrospun nanofibers can be achieved through the simple blending of the polymer solution before spinning, post-spinning surface functionalization methods, or using core-shell electrospinning techniques [7]. The choice of the method is dependent on the intended application and based on the preferred bioactive agent release profile.

Herein, crude CA extract, a medicinal plant, was selected to be incorporated into the PVA_CS-TPP nanofibers to improve the healing properties of the double-layered nanocomposites membranes. The cumulative release profile of the crude CA extract from the bottom’s nanofibers is shown in Figure 5. The amount of CA released from the bottom layer reached 84.22 ± 2.08% after 10 days. The slower and sustained release of CA could be influenced by the nature of the plant components as well as the pore size of the nanofibers. Further, the swelling ability and consequential diffusion and degradation of the polymers on exposure to the aqueous medium (PBS pH = 5.5) can also affect the release mechanism of the crude CA extract [62,63].

Regarding all the above-mentioned, the obtained release profile is crucial for the produced bottom layer to prevent wound bacterial colonization and infection.

### 3.4. Assessment of the Antibacterial Properties of the Produced Electrospun Double-Layered Nanocomposites Membranes

Wounded skin is usually associated with a higher level of bacterial colonization and subsequent biofilm formation, which can prolong the inflammatory phase of healing, delay collagen synthesis, and impede re-epithelialization [1,5,6]. Although the inflammatory cells inducing phagocytosis of the pathogens, high growth of bacteria at the wound site lead to an extended infiltration of immune cells and increase the production of pro-inflammatory cytokines [1,5,6]. Therefore, an ideal wound dressing material should provide an unfavorable environment for bacteria growth or inhibit the infiltration of bacteria at the wound site.

In this study, the behavior of the PCL’s top layer was similar to the control group (filter paper with a pore size of 0.22 µm), proving the capability to act as a protective barrier against *S. aureus* and *P. aeruginosa* due to its low porosity, Figure 6. The antibacterial activity of the bottom layers (PVA_CS-TPP and PVA_CS-TPP_CA) was also evaluated. PVA_CS-TPP showed an inhibitory effect of 61.80 ± 4.45% and 48.41 ± 9.94% against *S. aureus* and *P. aeruginosa*, respectively. However, an increased inhibitory effect on bacterial growth, 99.96 ± 6.04% and 99.94 ± 0.67% for *S. aureus* and *P. aeruginosa,* was observed when the crude CA extract was incorporated, Figure 6. This result confirmed that the CA’s secondary metabolites, like triterpenoid saponins (asiatic acid, asiaticoside, and madecassoside) exhibit a high ability to inhibit bacterial growth, as previously reported in different research studies [34,64]. Hence, the PVA_CS-TPP layer containing CA extract is essential to provide an aseptic environment at the wound site.

### 3.5. Analysis of the In Vitro Cell Viability

In this study, the effect of the produced double-layered nanocomposites membranes on cell viability and proliferation was investigated through MTT assay for 1, 3, and 7 days and the results are shown in Figure 7. Overall, it was found that the PCL/PVA_CS-TPP and PCL/PVA_CS-TPP containing CA extract displayed excellent cell viability (over 90%) when tested on NHDF cells. The non-toxic properties could be attributed to the porous and hydrophilic character exhibited by the bottom layers, which can support cell adhesion and proliferation.

Similarly, Yousefi et al. [35] demonstrated that the incorporation of the Henna extracts into CS/PEO nanofibrous mats did not affect the proliferation and viability of the NHF cells, confirming that they lack cytotoxic effects.

## 4. Conclusions

In the present study, the electrospun double-layered nanocomposite membrane of PCL, PVA, and CS-TPP was incorporated with crude CA extract to improve skin regeneration. The obtained nanocomposite material was characterized in terms of physical, chemical, and biological features to verify its suitability to be applied as an advanced wound dressing. The results showed that the PCL’s layer could act as a protective barrier against external contaminants, and simultaneously, the PVA_CS-TPP_CA’s bottom layer was revealed to be effective in exudate absorption and providing a moist environment. Moreover, PVA_CS-TPP_CA improved the antibacterial activity of the double-layered nanocomposite membrane against *S. aureus* and *P. aeruginosa*, which is advantageous to prevent microbial penetration. This novel nano-carrier was also able to achieve a controlled release of CA and closely resemble the mechanical properties of native skin.

Furthermore, the produced double-layered nanocomposites membranes did not provide a cytotoxic effect on NHDF cell culture. Hence, these findings emphasize the potential to blend PCL and PVA_CS-TPP nanofibers with the beneficial properties of crude CA extract for wound healing applications.

Shortly, other potential healing benefits of these double-layered nanocomposites membranes will be further tested, as well as their efficacy in animal models to establish in vivo data.

## Figures and Tables

**Figure 1 polymers-12-02653-f001:**
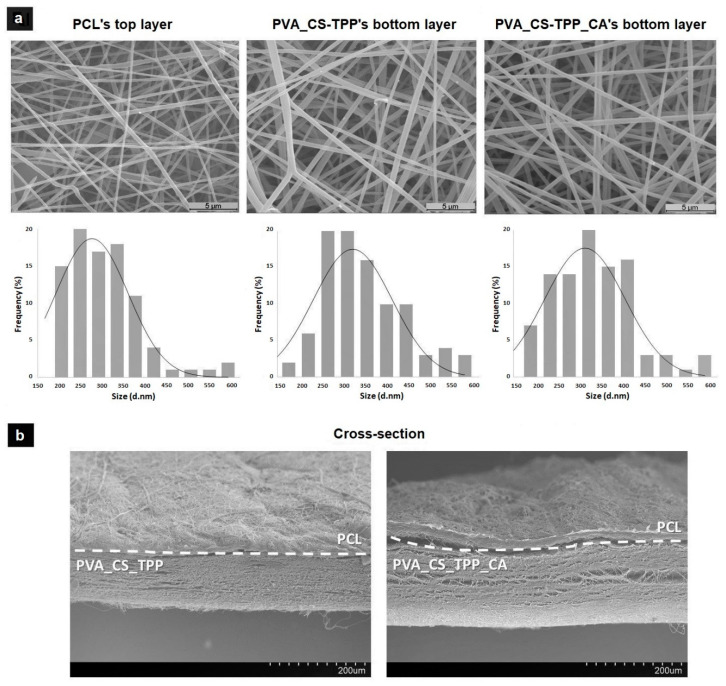
Morphology and fiber diameter distribution of the Polycaprolactone (PCL)’s top layer and the Poly(vinyl alcohol) (PVA) and Chitosan-Sodium tripolyphosphate (CS-TPP) (PVA_CS-TPP) and PVA_CS-TPP containing *Centella asiatica* (L.) (CA)’s bottom layers (**a**); and cross-sectional SEM images of the double-layered membranes; (**b**) PCL/PVA_CS-TPP with crude CA extract (on the right) and without (on the left).

**Figure 2 polymers-12-02653-f002:**
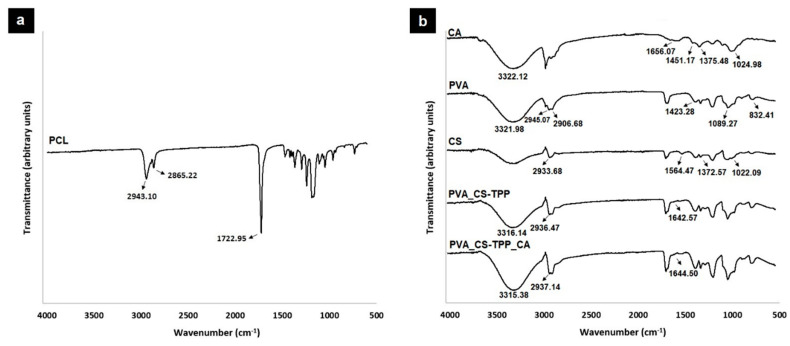
ATR-FTIR analysis of the produced double-layered membranes. FTIR spectra of the PCL’s top layer (**a**), the bottom layers of PVA_CS-TPP and PVA_CS-TPP_CA, and their raw materials (**b**).

**Figure 3 polymers-12-02653-f003:**
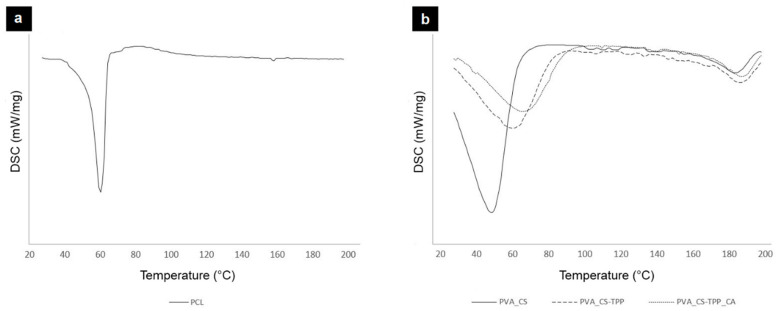
Characterization of the thermal behavior of the produced double-layered membranes. DSC curves of the PCL’s top layer (**a**); and the bottom layers of PVA_CS, PVA_CS-TPP, and PVA_CS-TPP_CA (**b**).

**Figure 4 polymers-12-02653-f004:**
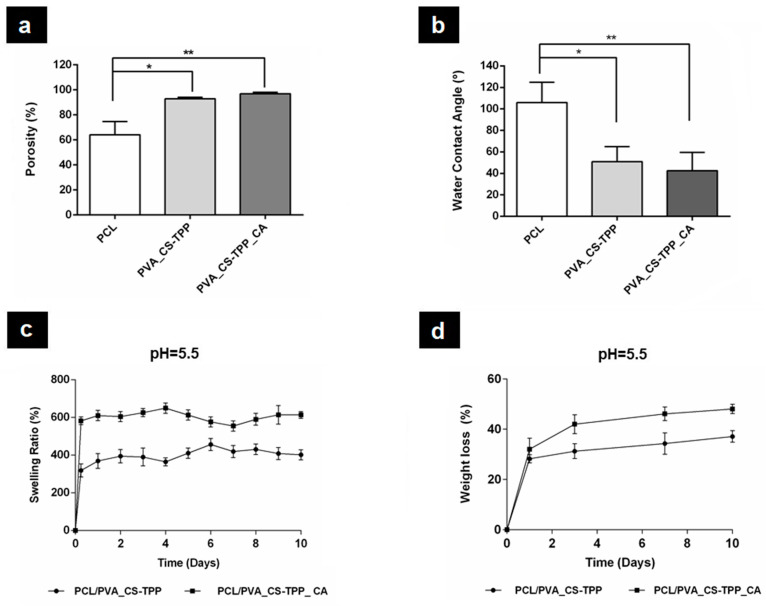
Characterization of the total porosity (**a**); wettability (**b**); swelling profile (**c**); and weight loss at pH = 5.5 (**d**) of the produced double-layered membranes. (Data are represented as average ± standard deviation (S.D.), * *p* < 0.05 and ** *p* < 0.001).

**Figure 5 polymers-12-02653-f005:**
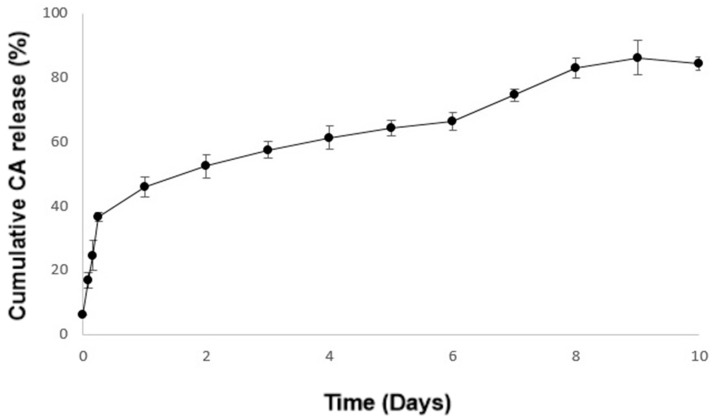
Cumulative release profile of the crude CA extract from the bottom’s PVA_CS-TPP nanofibers at pH = 5.5.

**Figure 6 polymers-12-02653-f006:**
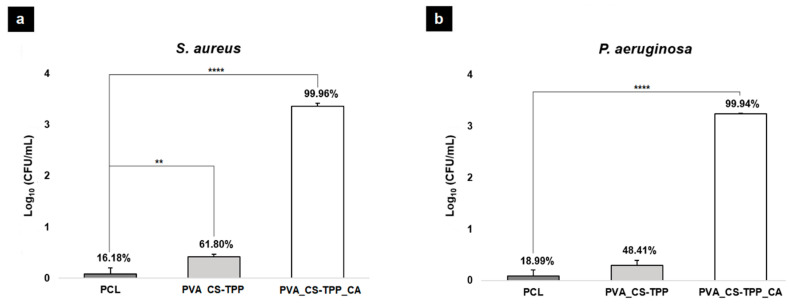
Evaluation of the antibacterial potential of the PCL’s top layer and the PVA_CS-TPP_CA and PVA_CS-TPP_CA’s bottom layers against *Staphylococcus*
*aureus* (*S. aureus*) (**a**) and *Pseudomonas aeruginosa* (*P. aeruginosa*) (**b**). (Data are represented as average ± standard deviation (S.D.), ** *p* < 0.001 and **** *p* < 0.0001).

**Figure 7 polymers-12-02653-f007:**
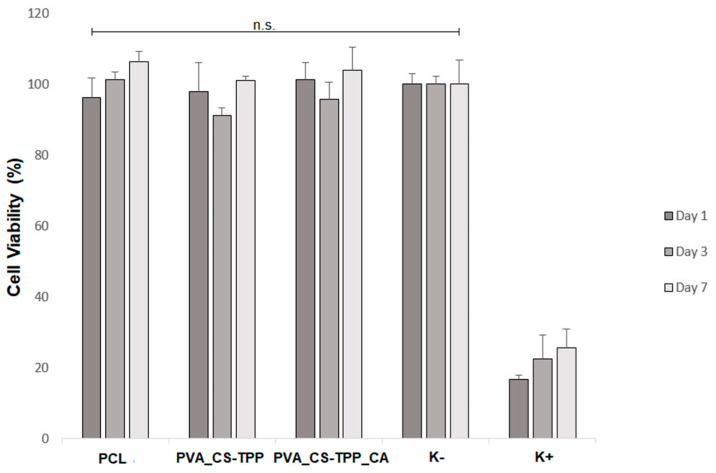
Growth of normal human dermal fibroblasts (NHDF) cells on the PCL’s top layer and the PVA_CS-TPP and PVA_CS-TPP_CA’s bottom layers detected by 3-(4,5-Dimethyl-2-thiazolyl)-2,5-diphenyl-2H-tetrazolium bromide (MTT) assay in term of cell viability after 1, 3, and 7 days.

**Table 1 polymers-12-02653-t001:** Evaluation of the mechanical properties of the produced Polycaprolactone (PCL)/ Poly(vinyl alcohol) (PVA) and Chitosan-Sodium tripolyphosphate (CS-TPP) (PVA_CS-TPP) nanofibrous membranes with and without crude *Centella asiatica* (L.) (CA) extract and comparison with the native human skin values.

	Tensile Strength (MPa)	Young’s Modulus (MPa)	Elongation at Break (%)	Thickness (mm)
**PCL/PVA_CS-TPP**	3.96 ± 0.99	38.16 ± 5.32	10.39 ± 1.89	0.12 ± 0.01
**PCL/PVA_CS-TPP_CA**	3.03 ± 0.67	36.36 ± 7.29	8.31 ± 0.61	0.10 ± 0.02
**Native skin**	2.50–30.00 ^a^	0.40–20.00 ^a^	10.00–115.00 ^a^	−

^a^ Native skin values were obtained from reference [10].

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
