# Peer review of "Preparation and Characterization of Electrospun Double-layered Nanocomposites Membranes as a Carrier for *Centella asiatica* (L.)"

_polymers, 2020, doi:10.3390/polym12112653_

Round 1

Reviewer 1 Report

The present manuscript deals with the preparation and characterization of a double-layered nanocomposite membrane as wound dressing. 

Some points should be addressed:

In the abstract, keywords, introduction and along the manuscript it is not clear if the abbreviation TPP refers to tyamine pyrophosphate or sodium tripolyphosphate. Please amend it.

A characterization in terms of qualitative and quantitative chemical composition of the ethanolic solution derived from maceration of Centella Asiatica is required. Consider also, if the terminology extract is correct in this context.

Line 286 As a positive control, have been tested cells treated with EtOH (96%)?

Line 313-314 A more detailed description of Figure 1b is required

Line 347-357 The comments about DSC traces of bottom layer (Figure 3b) are not clear. Actually, I do not observed any glass transition, since glass transition is not revealed by DSC as an endothermic event as stated by the authors in line 348. Please revised all paragraph 3.2.3.

Line 510 The authors have not demostrated that the proposed membrane provides "nutrient supply and gas exchange". 

Reviewer 2 Report

The manuscript reports a double-layer electrospun nanofiber films. One layer contained Chitosan-Thiamine Pyrophosphate (CS-TPP) and Poly(vinylalcohol) (PVA) foe delivering Centella asiatica (L.) (CA). The other layer was composed of Polycaprolactone (PCL) nanofibers, which had antibacterial activity against Staphylococcus aureus (S. aureus) and Pseudomonas aeruginosa (P. aeruginosa) because of avoiding bacterial penetration while endowing bioactive properties. These contents are interesting and should be welcome by the readers of POLYMERS. I recommend its acceptance for publications after the following issues are well addressed. 1)The title can be “Preparation and Characterization of Electrospun Double-layered nanocomposites membranes as a carrier for Centella asiatica (L.)”. 1)The INTRODUCTION should give a full background about the development of electrospinning, from which the merits of your job can be further projected, e.g. electrospinning is fast developing from traditional blending process (https://doi.org/10.1016/j.polymertesting.2020.106872) to novel single-fluid (Polymers, 2020, 12, 2421), coaxial (10.1016/j.msec.2020.110988), modified coaxial (10.1016/j.ijpharm.2020.119397), tri-axial (Polymers 2020, 12, 2034), side-by-side (Polymers 2020, 12, 2413), and other multiple-fluid processes (10.1016/j.jallcom.2020.156471) for creating core-shell (10.1016/j.ijbiomac.2020.02.239), Janus (10.1016/j.matdes.2020.109075) and other complicated nanostructures (10.1016/j.matdes.2020.108782). However, those new methods for manipulating the inner chamber structure are complicated. Thus, the traditional single-fluid blending electrospinning is still the mainstream because it is facile to conduct, easy to be scaled up, and powerful for tailoring the components and compositions of resultant composite nanofibers. What is more, the manipulation of deposition on collector with different kinds of electrospun nanofibers can be further explored to improve the functional performances of the final products. Your present work is just a fine example of this kind. 2)In the section of “1.4” about fabrication, the information about the fluid flow rate is neglected. 3)In the in vitro dissolution tests, how much was the volume of the dissolution media? What is the solubility of CA, does the in vitro tests meet the sinking condition? 4)In Figure 2, the FTIR spectra of CA, PVA and CS should also provided for comparison. 5)The reverences’ formats should be unified according to the journal’s request, paying attention to the upper case or lower case in the article title, e.g. Ref. [16]. 6)In order to maintain active ongoing threads of scientific discussion, the authors are encouraged to demonstrate the connection between the work and recent reports published in Polymers, which should help to increase the impact of your work after publication.

Round 2

Reviewer 1 Report

The manuscript has been improved and ready for pubblication

Reviewer 2 Report

The manuscript's quality has been substantially improved after revision.

Their responses are clear and convinced. 

I recommend its acceptance for publication in its present form.